# The monsoon-associated equine South African pointy mosquito '*Aedes caballus*'; the first comprehensive record from southeastern Iran with a description of ecological, morphological, and molecular aspects

Jalil Nejati[1], Shahyad Azari-Hamidian[2], Mohammad Ali Oshaghi [3]*, Hassan Vatandoost[3], Vanessa L. White[4], Seyed H. Moosa-Kazemi[3], Rubén Bueno-Marí[5,6], Ahmad A. Hanafi-Bojd[3], Nancy M. Endersby-Harshman[4], Jason K. Axford[4], Fateh Karimian[7], Mona Koosha[3], Nayyereh Choubdar[3], Ary A. Hoffmann[4]

1 Health Promotion Research Center, Zahedan University of Medical Sciences, Zahedan, Iran, 2 Research Center of Health and Environment, School of Health, Guilan University of Medical Sciences, Rasht, Iran, 3 Department of Medical Entomology and Vector Control, School of Public Health, Tehran University of Medical Sciences, Tehran, Iran, 4 Bio21 Institute, Pest and Environmental Adaptation Group, School of BioSciences, The University of Melbourne, Victoria, Australia, 5 Departamento de Investigación y Desarrollo (I+D), Laboratorios Lokímica, Valencia, Spain, 6 Parasites & Health Group, Department of Pharmacy, Pharmaceutical Technology and Parasitology, Faculty of Pharmacy, University of Valencia, Valencia, Spain, 7 Department of Parasitology, Pasteur Institute of Iran, Tehran, Iran

* moshaghi@sina.tums.ac.ir

## Abstract

The equine South African pointy vector mosquito, *Aedes caballus*, poses a significant threat to human health due to its capacity for transmitting arboviruses. Despite favorable climate for its existence in southeast Iran, previous records of this species in the area have indicated very low abundance. This comprehensive field and laboratory study aimed to assess its current adult population status in this region, utilizing a combination of ecological, morphological and molecular techniques. Four distinct types of traps were strategically placed in three fixed and two variable mosquito sampling sites in the southern strip of Sistan and Baluchistan Province. Subsequently, DNA was extracted from trapped mosquitoes and subjected to PCR amplification using the molecular markers COI, ITS2, and ANT. In total, 1734 adult *Ae. caballus* specimens were collected from rural areas, with the majority being captured by $CO_2$-baited bednet traps. A notable increase in the abundance of this species was observed following rainfall in February. The genetic analysis revealed multiple haplotypes based on COI and ITS2 sequences, with COI gene divergence at 0.89%, and ITS2 sequence divergence at 1.6%. This suggests that previous challenges in morphological identification may have led to misidentifications, with many adults previously classified as *Ae. vexans* potentially being *Ae. caballus*. The findings of this study hold significant implications for public health authorities, providing valuable insights for integrated and targeted vector control and disease management efforts.

**Data Availability Statement:** All relevant data are within the paper and its Supporting information files.

**Funding:** Tehran University of Medical Sciences (Grant Number 9121260011). It was also supported by NHMRC Fellowship to Ary A. Hoffmann. The funders had no role in study design, data collection and analysis, decision to publish, or preparation of the manuscript.

**Competing interests:** The authors have declared that no competing interests exist.

## Introduction

Mosquitoes (Diptera: Culicidae) are the most important insects from a public health point of perspective due to their role in transmitting debilitating human diseases such as malaria, dengue, encephalitis, and filariasis [1]. The family Culicidae comprises two subfamilies: Anophelinae and Culicinae, encompassing 41 or 113 genera, depending on the classification of the tribe Aedini, with a total of 3,719 species [2].

The genus *Aedes* belongs to the tribe Aedini, the largest within the family, and includes 1296 recognized species [3]. In recent years, a notable change in mosquito classification has been attempted elevate many subgenera to generic rank within the tribe Aedini based on phylogenetic analyses. This resulted 82 genera of Aedini, most of which were previously documented as the subgenera of *Aedes* [4–7]. In the traditional classification, it was decided that the tribe includes 10 genera and *Aedes* should be a distinct genus with several subgenera [8]. However, this reclassification has been met with controversy, and the issue still remains unresolved [3]. Given this uncertainty, two common classifications mentioned in published papers are Reinert et al and Wilkerson et al. [7–9].

In the genus *Aedes*, *Ae.* (*Aedimorphus*) *vexans* (Meigen, 1830), known as the flood mosquito, has been documented a vector of microfilariae of causing agents of dirofilariasis, setariasis, and Rift Valley fever and West Nile fever viruses [10–13]. This species is primarily distributed in the Holarctic and Oriental regions, with minor extensions into the Australasian Region (excluding Australia). It also occurs in the Afrotropical Region, as well as in Central America and South Africa [1]. Research conducted in the United States found that this species displayed notably increased feeding activity in prairie and meadow habitats compared to forested areas. Their attack rates peaked sharply around 30 minutes after sunset [14]. Interestingly, the level of light by itself did not have a significant impact on their activity; attacks occurred even in bright sunlight, as long as the temperature conditions were suitable. However, the notable surge in activity observed after sunset suggested a stimulating effect caused by reduced illumination. This mosquito remained active within a temperature range of 10–34°C [15]. The ideal temperature for *Ae. vexans* development is 30°C, resulting in a 6–8-day transition from larval hatching to adult emergence [16]. Various types of light traps are commonly used as preferred sampling tools for this species from dusk to dawn, resulting in a higher collection of specimens. Additionally, other traps such as resting box, carbon dioxide ($CO_2$)-baited traps have been employed [14, 17–19].

Another *Aedes* species, *Ae. (Ochlerotatus) caballus* (Theobald, 1912) [*Ochlerotatus* (*Juppius*) *caballus*], is considered a vector of Rift Valley fever, Wesselsbron, Middelburg and West Nile viruses [20–23]. It is known an Afrotropical species [24]. In the Middle East it has been recorded from Iran, Saudi Arabia and Yemen [25]. This species typically follows a single-generation breeding pattern and is rarely found indoors, preferring to stay near its breeding sites [26]. It lays eggs singly on soil, between rocks, near rivers, or bodies of water [24]. This mosquito breeds abundantly in small to medium-sized breeding habitats with vegetation, often filled periodically by rain or irrigation water. The occurrence of Rift Valley fever epidemics, coinciding with rainfall patterns, is aligned with its breeding cycle. Notably, not all eggs hatch simultaneously, which enhances their chances of survival. These mosquitoes emerge in large swarms near their breeding sites and are known for their painful bites. They may settle on humans and clothing, immediately engaging in biting. Humans, and some mammals like horses, cattle, and sheep serve as hosts [26]. In a study conducted in South Africa, different $CO_2$ trap types were used to sample it [27]. Although it has been mentioned as a threat to health, its ecological (field observations and adult mosquito collection for sending to the laboratory), morphological, and molecular aspects have rarely been investigated.

At least 71 mosquito species representing 8 genera have been documented in Iran, including 13 species of the tribe Aedini [1, 28, 29]. The southeast of the country is affected by the periodic monsoon systems that can cause occasional heavy precipitation [30, 31]. Invasion, nuisance, and biting of mosquitoes after rainfall is considered an important health issue in Chabahar and Konarak Counties. Previously, *Ae. vexans* was recorded as the dominant adult *Aedes* species in this region, while *Ae. caballus* had a significantly lower population density [32, 33]. It appears that the numerous and diverse water bodies following monsoon rainfalls can create a suitable climatic environment for the increased density of *Ae. caballus* in the southeast of Iran. The coexistence of larval habitats for this species and *Ae. vexans* has been reported in the southern part of the country, with a significant affinity index [34]. This raises our expectation for a larger population of *Ae. caballus* in Chabahar and Konarak Counties. Given the morphological similarities between these species, there is potential for misidentification, highlighting the importance of careful morphological investigation.

*Aedes caballus*, *Ae. chelli* (Edwards, 1915), and *Ae. juppi* (McIntosh, 1973) are three morphologically similar species within the subgenus *Ochlerotatus* [1, 21]. Barcode sequencing can help confirm their morphological identifications. In South Africa, the cytochrome oxidase [35] gene was amplified for the identification of *Ae. caballus* and *Ae. juppi* [27]. Additionally, DNA-based technology serves as a valuable tool for molecular identification and the establishing relationships among species [36, 37].

This field and laboratory study aimed to investigate ecological, morphological and molecular aspects of *Ae. caballus* in the southeast of Iran and also clarify its abundance in comparison to *Ae. vexans*.

## Materials and methods

### Study area

This study was conducted in Chabahar and Kanarak, two ports, in the southern strip of Sistan and Baluchistan Province, located in the southeast corner of Iran (Fig 1). Chabahar County (25.2966˚N, 60.6458˚E), located in southeastern Iran, shares its eastern border with Pakistan and is near the Oman Sea in the south. It is close to the Indian Ocean, which causes summer rainfalls due to the Monsoon Currents. This makes it the coolest and the warmest Southport in Iran during summer and winter respectively. On average, the highest temperature is around 34˚C, and the lowest is about 21.5˚C. The humidity levels range from 55% to 75%. Annual rainfall varies but averages around 150 mm. Chabahar is situated at a low elevation, with its highest point only about 45 meters above sea level [38, 39]. Konarak (25.3631˚N, 60.4001˚E) with an altitude of 5 meters above sea level (MASL) is another coastal county located along the shoreline of the Oman Sea, west of Chabahar. Its proximity to the sea, positioned along the path of the Indian subcontinent's monsoon winds, results in a temperate tropical climate with high relative humidity. The county often experiences significant rainfall and brief but intense storms, which are notable climatic features [40, 41]. In this region, numerous water bodies (wetlands/ponds replenished by rain) as well as open cement water reservoirs, serve as the main mosquito breeding habitats in rural areas. Date palm and banana gardens, small farms, and even paddy fields in these counties can support the presence, establishment, and breeding of mosquitoes due to their history of malaria and other vector-borne diseases [42]. Based on the previous modeling and field investigations, both their urban and rural areas were selected for this study [43].

In this study, one urban and two rural areas were selected as three fixed sites for adult mosquito collection: Konarak (an urban area), Paroomi (25.443091˚N, 60.906849˚E, 44 MASL), and Vashname-Dori (25.371683˚N, 60.807573˚E, 19 MASL), two villages belonging to

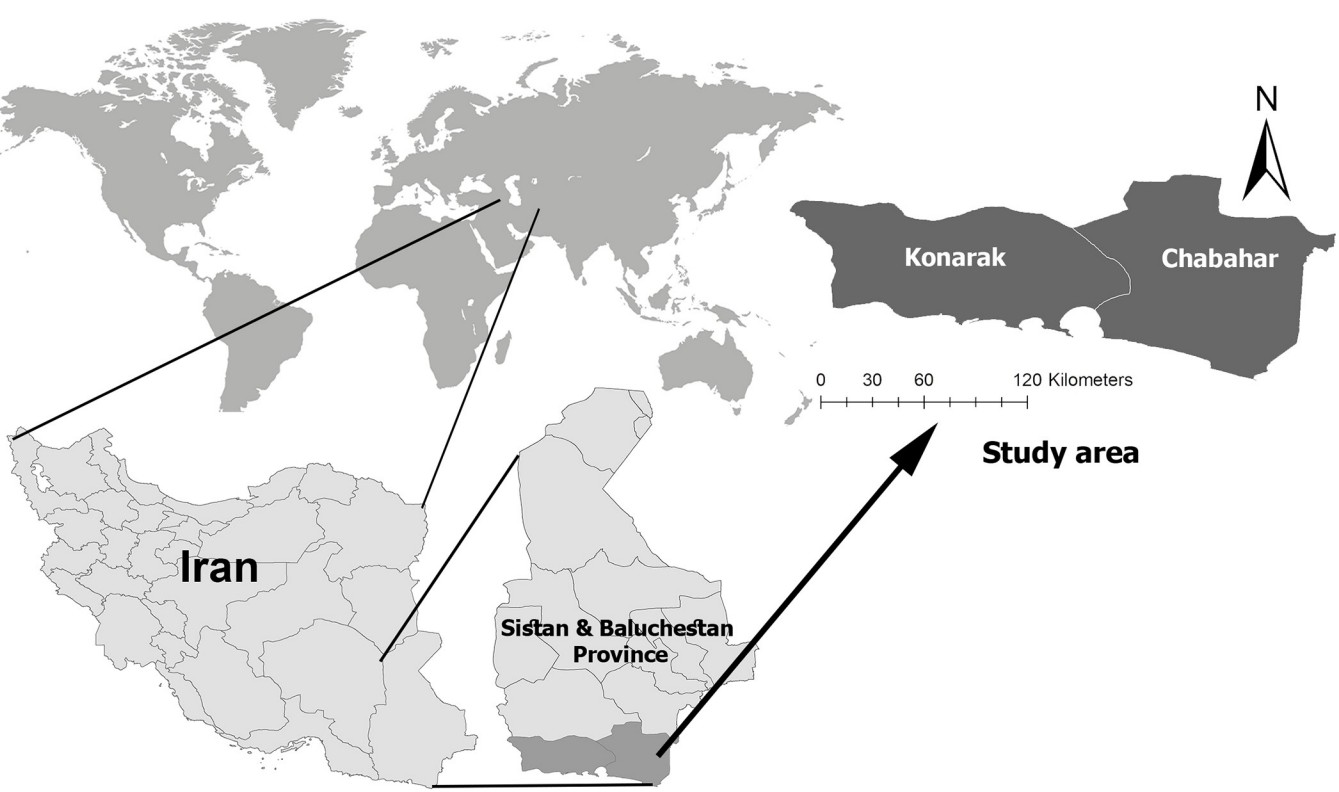

**Fig 1. Geographic location of the study area.** Prepared using ESRI ArcGIS 10.3 and Adobe Photoshop CS6 softwares.

Chabahar. Sporadic collections were also made in two additional Chabahar villages: Nalint (25.762432˚N, 61.417853˚E, 43 MASl) and Kachoo (25.244452˚N, 60.899460˚E, 17 MASL).

### Field sampling and morphological identification

In the present study, various trap types were employed to collect *Ae. caballus*, including $CO_2$-baited bednet traps, resting box traps, malaise traps and BG-Sentinel traps (Fig 2). The sampling schedule was relatively consistent with a previous study conducted in the same area, resulting in the collection of similar species. Sampling was conducted monthly from July 2016 to June 2017, starting before noon and continuing until one hour after dusk. Sample collection ranged from approximately 12 a.m. to 9 p.m. in spring and summer and 11 a.m. to 6 p.m. in autumn and winter.

At each sampling site, one trap of each type was positioned 50–100 meters apart from another trap. These traps were placed outdoors and checked every 15 minutes, except BG-Sentinel trap, which was monitored after dusk. The specimens collected were identified using an Iranian mosquito key which is based on the available types stored in the Natural History Museum (London) [44] and incorporates morphological characters from previous keys [45–47] The key includes many additional characters, taxonomic notes and data for certain taxa that aid in identification. Identified specimens were used for molecular studies.

### Laboratory study

Molecular analyses on collected adult mosquitoes were performed in both Australia (Bio21 Institute, The University of Melbourne) and Iran (Insect Molecular Laboratory of Department

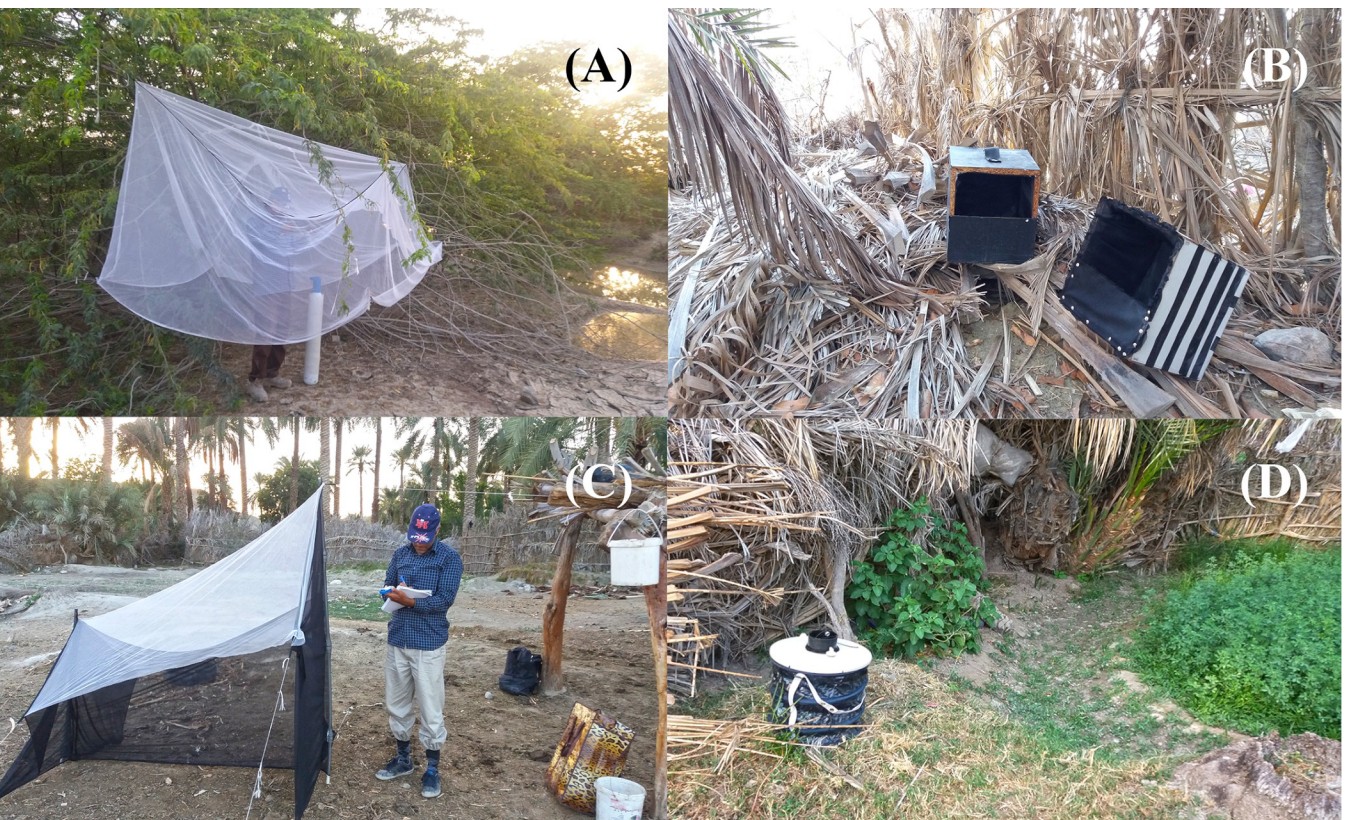

**Fig 2. Adult collection traps of *Aedes caballus* and *Ae. vexans*. (A)** CO2-baited bednet trap. **(B)** Resting box trap. **(C)** Malaise trap. **(D)** BG-Sentinel trap. Original photos by Jalil Nejati.

of Medical Entomology and Vector Control, Tehran University of Medical Sciences). DNA extraction was carried out by three methods: Chelex[R], Collins [48] and G-spin™ kit. In the Chelex[R] method, DNA was extracted from each sample crushing it with two glass beads (3 mm) in 250 µL of 5% Chelex [R] (Bio-Rad Laboratories) and 3 µL proteinase K solutions. Samples were homogenized in a TissueLyser (Qiagen, Hilden, Germany) for 1 min at 25 Hz. After centrifuging, they were incubated in a 65°C water bath for 30 min. To inactivate the proteinase K, they were boiled for 10 min in a 90°C water bath. The homogenates (as PCR templates) were temporarily refrigerated at 4°C and then transferred to -20°C freezers for long-term stability. Before PCR, tubes were centrifuged at 14,000 rpm for two minutes. The aqueous DNA was pipetted from just above the Chelex[R] resin, ensuring the resin remained in the tube. A solution of 1:10 or 1: 3 (depending on PCR performance) was used in PCR. In Iran, some samples comprised whole bodies of dehydrated mosquitoes that were crushed using an autoclaved glass pestle in 1.5 mL micro tubes. DNA of individual dried mosquitoes was extracted using Collins' extraction method and stored at -20°C until use. Additionally, it was extracted using the G-spin™ Genomic DNA Extraction Kit (iNtRON Biotechnology, Korea), following the manufacturer protocol.

The extracted DNA was used for PCR amplification with three diverse molecular markers and sequencing: mitochondrial cytochrome oxidase subunit I gene, the internal transcribed spacer 2 (ITS2) of ribosomal RNA gens, and the nuclear adenine nucleotide translocase gene (ANT) (Table 1) as described previously [35, 49–52].

**Table 1. Primers used in to amplify diagnostic DNA regions for *Aedes caballus*.**

| Target gene/locus | Primer name | sequence (5'-3') | Product size (bp) |
|---|---|---|---|
| COI | FCOI | GGTCAACAAATCATAAAGATA | 710 |
| | RCOI | TAAACTTCAGGGTGACCAAAAAATCA | |
| ITS2 | 5.8s | TGTGAACTGCAGGACACATGAA | 454–484 |
| | 28s | ATGCTTAAATTAGGGGGTAGTC | |
| ANT | FANT | TGCTTCGTNTACCCVCTKGACTTTGC | 290–450 |
| | RANT | CCAGACTGCATCATCATKCGRCGDC | |

PCR was carried out in standard 30 μL reactions containing 5 μL diluted DNA, 3 μL (10×) PCR buffer with 2 mM $MgCl_2$, 2.9 μL dNTPs (2.5 mM), 1.2 μL of both forward and reverse primers (10 μM), 0.3 μL BSA, 0.3 μL Taq DNA polymerase (New England BioLabs) and 16.1 μL $ddH_2O$. In Iran, the ready mixture 'Taq DNA Polymerase Master Mix RED' (Ampli-qon, Denmark) was used. It included all the necessary materials for PCR as a mixture: Tris-HCl pH 8.5, $(NH_4)_2SO_4$, 2 mM $MgCl_2$, 0.2% Tween® 20, 0.4 mM dNTP, 0.2 units/μL Ampli-qon Taq DNA polymerase. Therefore, only primers, extracted DNA samples and double sterile distilled water ($ddH_2O$) were added.

PCR products were amplified using the following thermocycling parameters: for COI: primary denaturing was at 94˚ for 5 min, then 5 cycles of 94˚ for 40 s, 45˚ for 60 s, 47˚ for 1 min and then 35 cycles of 94˚ for 40 s, 51˚ for 60 s, 72˚ for 1 min, and final elongation at 72˚ for 5 min. For ITS2 program started with 94˚ for 5 min, then 35 cycles of 94˚ for 1 min, 58˚ for 1 min and 72˚ for 2 min, followed by 72˚ for 4 min. For ANT thermal program started with 95˚ for 4 min, then 35 cycles of 95˚ for 45 s, 59˚ for 45 s, 72˚ for 1 min followed by 72˚ for 4 min.

PCR products were visualized on a UV trans-illuminator or in Bio-Rad Gel Doc XR after electrophoresis in a 1% agarose gel (Agarose, MP, Sigma) containing ethidium bromide stain. Only PCR products of expected size (see Table 1) with high stain intensity and represented by single bands (one product only) were sequenced at Macrogen Inc. (Seoul, South Korea) for both the forward and reverse DNA strands using the primers described above.

The sequences were aligned and manually edited with Geneious 10.1.3 [53].

To ensure accuracy of the sequences, they were compared with entries in the GenBank using BlastN (Basic Local Alignment Search Tool, https://blast.ncbi.nlm.nih.gov). The Clustal Omega software was used to compare the sequences that were retrieved from GenBank with the current specimens as well as phylogenetic analysis. Consensus sequences for the regions described above were submitted to the GenBank database (see Table 2).

## Results

A total of 1734 adult *Ae. caballus* were collected, all from rural sites. The majority of mosquitoes were caught in $CO_2$-baited bednet traps (1031, equivalent to 59.46% of specimens) followed by malaise (698, 40.25%), BG-Sentinel 2 (4, 0.23%) and resting box (1, 0.06%) traps. An increase in the abundance of this species was observed coinciding with rainfall in February. No specimens were collected before and after this period (Fig 3).

### Morphological identification

Morphological characters relevant to identification of *Ae. caballus* were assessed in all collected specimens as follows: The hindtarsomeres 5 entirely dark, the lower mesepimeral setae present, and the wings profusely speckled with dark scales, while pale scales were present dorsobasally on

**Table 2. Details of the sequenced *Aedes caballus* specimens for COI, ITS2, and ANT region compared with GenBank entries.**

| Gene | Collection site | Code | Length (bp) and Accession. No. | Closest relatives according to BLASTn Species Length (bp) Identity % | Accession No. and origin |
|---|---|---|---|---|---|
| COI | Nalint | N5 | 678 MH709107 | *Ae. caspius* 674 96.14 | MT708649 Serbia |
| | Kachoo | KC1 | 669 MH634431 | *Ae. dorsalis* 673 96.11 | MG242488 USA |
| | | KC5 | 695 MH634432 | | |
| | Paroomi | P3 | 678 MH709108 | *Ae. spencerii* 658 95.74 | KF535007 Canada |
| | | P4 | 673 MH634433 | | |
| | Vashnam | V2 | 701 MH634434 | *Ae. caballus* 600 95.8 | MW077860 South Africa |
| | | V3 | 674 MH634435 | | |
| ITS2 | Nalint | A03 | 403 MN158185 | *Ae. sagax* 388 95.80 | KX866215 Australia |
| | | A06 | 398 MN158186 | | |
| | Paroomi | A20 | 398 MN158187 | *Ae. mallochi* 418 94.92 | KX866197 Australia |
| | Vashnam | A23 | 405 MN158188 | *Ae. scapularis* 388 92.75 | MT151958 Colombia |
| | | A25 | 403 MN158189 | | |
| ANT | Paroomi | P04 | 340 OM927722 | *Ae. albopictus* 1554 89.5 | XM_019673034 Predicted |
| | | P09 | 340 OM927723 | | |
| | Nalint | N15 | 340 OM927724 | *Cx.quinquefasciatus* 1487 88.5 | XM_038257177 Predicted |
| | | N16 | 340 OM927725 | | |
| | Vashnam | V17 | 340 OM927726 | *Ae.aegypti* 1442 88.2 | XM_021857589 Predicted |
| | | V21 | 340 OM927727 | | |

the costal, subcostal, and radial veins, abdominal terga without medially indented basal pale bands, as *Ae. vexans*, mostly paled scaled with special ornamentation. *Aedes vexans* collected from this region were distinguished from *Ae. caballus* by the presence of narrow basal pale bands on the hindtarsomeres (less than 0.25 length of metatarsomere), the absence of the lower mesepimeral setae, wing entirely dark-scaled, and slightly bilobed basal pale bands on abdominal terga. However, both species possessed visible long cerci, shared character of the aedine subgenera in Iran except *Stegomyia*, and proboscis longer than the forefemur. The proboscis is entirely dark in *Ae. caballus*, however have some pale scales in ventral surface in *Ae. vexans* (Fig 4).

## Molecular analyses

DNA analyses were carried out on 32 adult mosquitoes collected from the field. Clear electrophoretic images were obtained following PCR (Fig 5). The COI barcode region of seven *Ae.*

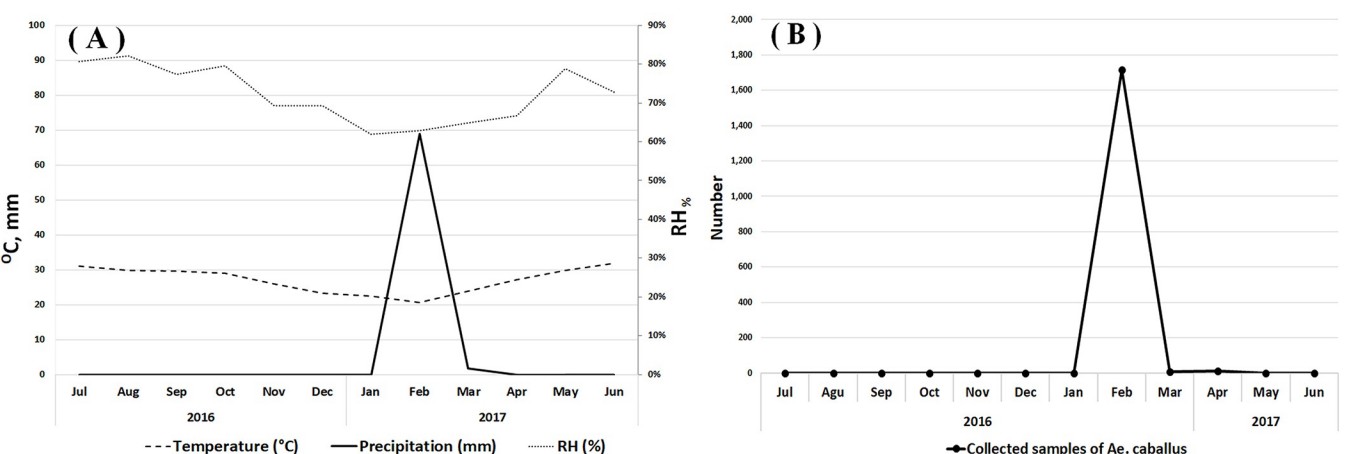

**Fig 3. Comparison of the meteorological variables and number of collected specimens. (A)** Monthly average temperature and humidity (fine lines) and rainfall (solid line); The weather data was obtained from the Sistan and Baluchistan Meteorological Organization. **(B)** Monthly abundance of *Aedes caballus* (collected only in February and April).

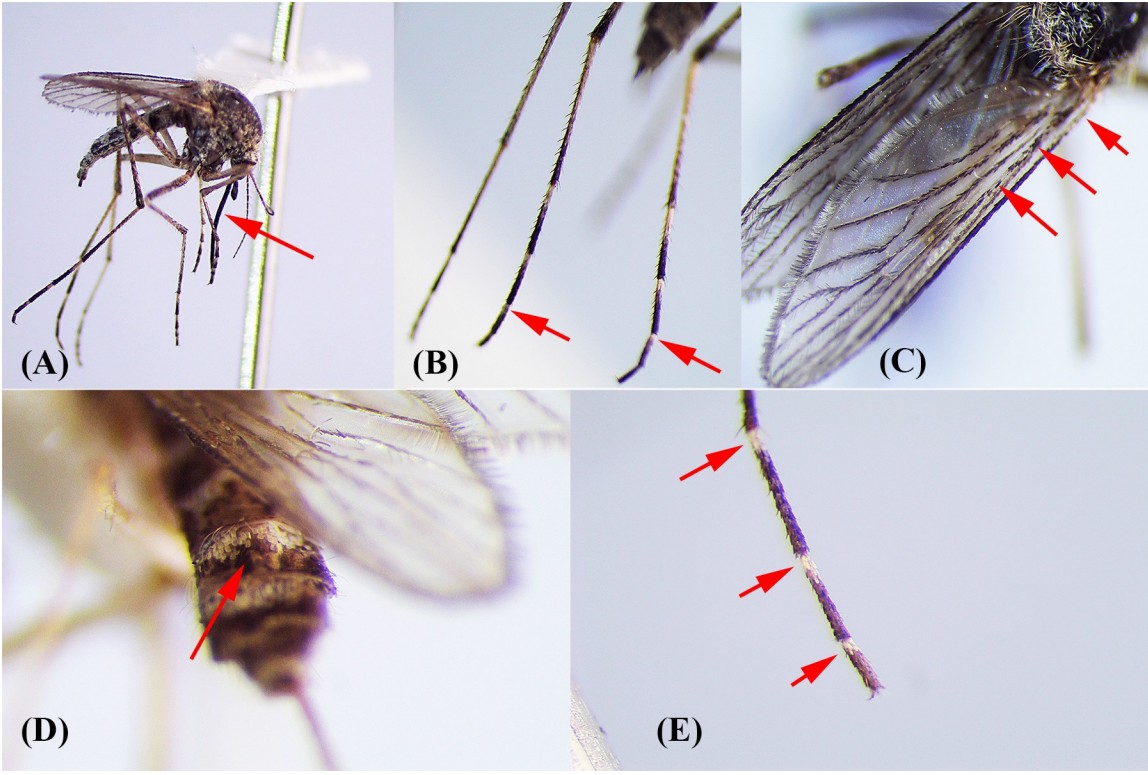

**Fig 4. Morphological characters of *Aedes caballus* (A-C) and *Ae. vexans* (D, E) collected from southeast of Iran. (A)** Proboscis length compared with forefemur. **(B)** Tarsomere 4 of all legs with basal pale band (no distinct), and hindtarsomere 5 entirely dark. **(C)** Speckled wings. **(D)** Bilobed basal pale bands, "V shape", in abdominal terga. **(E)** Narrow basal pale rings less than 0.25 length of tarsomere in hindtarsomeres (Original photos taken by Jalil Nejati).

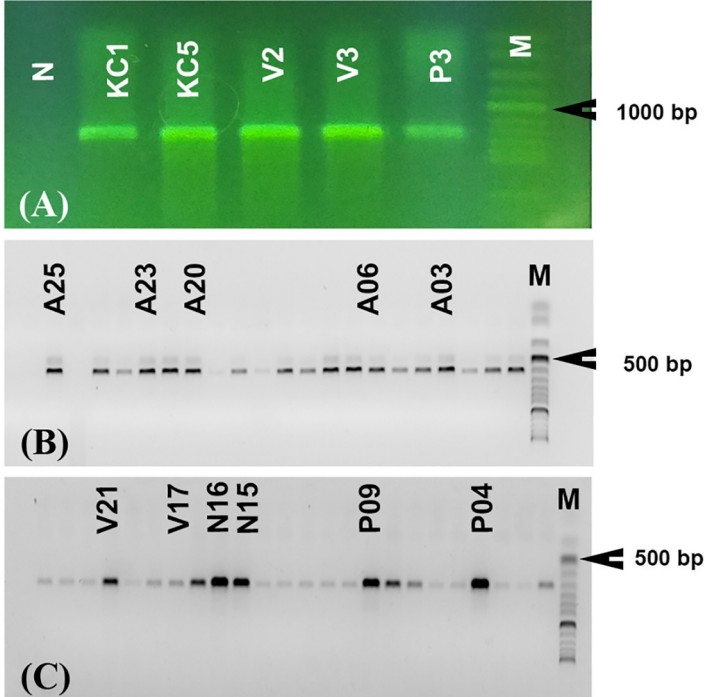

**Fig 5. Electrophoresis images showing PCR products generated by (A) COI, (B) ITS2, and ANT (C) sequences from *Aedes caballus* samples.**

*caballus* specimens were successfully amplified by the primers and protocols described above. PCR products used for sequencing consistently comprised a single band of expected size (673–701 bp) with high stain intensity (Fig 5). Consensus sequences for the described regions were submitted to the GenBank database (Table 2).

The comparison of the sequence homologies with the GenBank representative of *Ae. caballus* showed close similarity for COI (Table 2). The most similar sequences were *Ae. caspius*, *Ae. dorsalis*, *Ae. spencerii* and other *Ae. caballus* samples (all at around 96%) from diverse geographical regions. Notably, *Ae. vexans* was not included in the closest 100 specimens.

The Clustal Omega results revealed populations differed by up to six nucleotides in the COI region studied. The genetic diversity rate at 669 bp DNA sequences of the COI barcode region for different *Ae. caballus* populations was 0.89%, with 66.7% of the mutations being transitions and 33.3% were transversions (S1 Fig). The *Ae. caballus* COI sequence was aligned by Nucleotide BLAST. The sequence from Nalint showed the highest similarity to *Ae. caspius* with 674 bp and 96.14% identity.

A phylogenetic tree was generated using the Neighbor-Joining (NJ) method through MEGA7 [54]. The phylogenetic assemblages were made using current sequences along with four species with the highest percentage identity: *Ae. caspius*, *Ae. dorsalis*, *Ae. caballus* (South African sample) and *Ae. spencerii*. *Ae. vexans* and the outgroup *An. stephensi* were also included. Based on the phylogenetic tree, *Ae. caballus* specimens clustered in separate clades from the *Ae. vexans* or *Ae. caspius* specimens suggesting *Ae. caballus* is distant from the species. It is noteworthy that the Iranian *Ae. caballus* samples were not associated with the *Ae. caballus* specimens from South Africa (Fig 6).

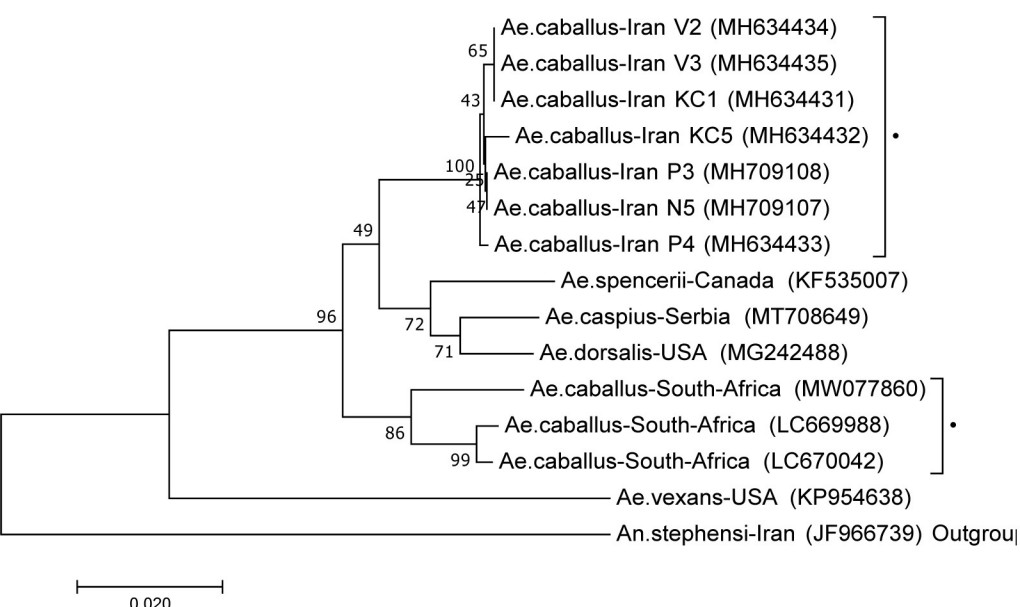

**Fig 6. Neighbour-joining tree, based on 669 bp DNA sequences of mtDNA COI barcode region, created using MEGA7.** *Aedes caballus* specimens from Iran (▲) and South Africa (▼) are grouped in two different clades. *Aedes vexans* is in a segregated branch. *Anopheles stephensi* used as an outgroup. Bootstrap values are shown on nodes. Scale bar shows genetic distance. Genbank IDs are shown on branches. Data were bootstrapped 500 times.

In total, 19 specimens of *Ae. caballus* were sampled for ITS2 analysis. PCRs were performed to amplify an approximately 430 bp fragment of this nuclear ribosomal gene. BLAST analysis showed the highest similarity (92.8–95.8%) to *Ae. sagax*, *Ae. mallochi*, and *Ae. scapularis* (Table 2).

Genetic diversity across the 365 bp sequences of the ITS2 locus for different populations of *Ae. caballus* was 1.6%. Six mutations including two transitions and four transversion were noted (S2 Fig). The most similar ITS2 sequences to *Ae. caballus*, in addition, *Ae. vexans* and *An. stephensi* (as outgroup) were used in a phylogenetic analysis which showed that *Ae. caballus* specimens were like each other, grouped into their own sub-clade. As in the COI phylogenetic tree, *Ae. caballus* was placed in a separate branch from *Ae. vexans* (Fig 7).

A 340 bp fragment of ANT was amplified for six samples of *Ae. caballus*. Unlike COI and ITS2, there were no variable nucleotide sites observed among the *Ae. caballus* samples from Iran. The closest similarities to the GenBank sequences were for *Ae. albopictus*, *Ae. aegypti* and *Cx. quinquefasciatus* (Table 2). Only the former 2 were included in the phylogenetic analysis along with the outgroup *An. stephensi*. The phylogenetic tree grouped the *Ae. caballus* Iran specimens into one clade (Fig 8) and these were separated from all other species with high bootstrap support including *Ae. caspius* from Iran (98%).

For all three markers, the outgroup *An. stephensi* was the most genetically divergent of all the species, as expected.

## Discussion

While the Afrotropical species *Ae. caballus* is known as the vector of Rift Valley fever, Wesselsbron and Cell fusing agent viruses, and other agent diseases [26, 27], research on this species is limited.

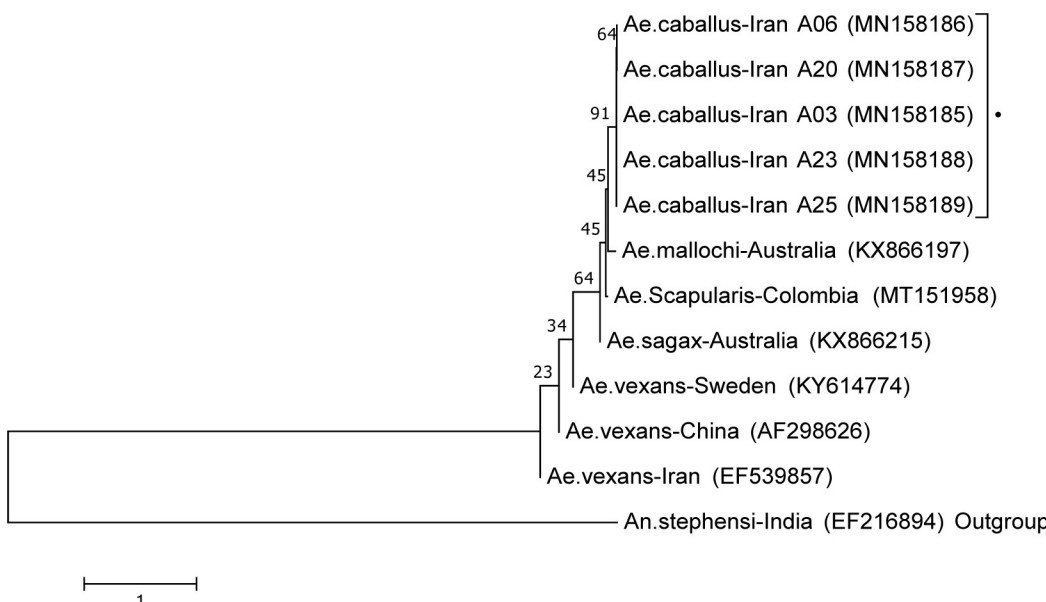

**Fig 7. Neighbour-joining tree, based on 403 bp DNA sequences of ITS2 rDNA region, created using MEGA7.** *Aedes caballus* from Iran (bold triangle ▲) grouped in one clade. *Anopheles stephensi* used as an outgroup. Bootstrap values are shown on nodes. Scale bar shows genetic distance. Genbank IDs are shown on branches. Data were bootstrapped 500 times.

Discovery of *Ae. caballus* in Djask, Hormozgan Province, southern Iran, was recorded for the first time by Edwards in 1935 [55]. Subsequent reports in the following years included only a limited number of larvae from other cities in that province, such as Minab and Beshagard [24, 34]. In 1983, *Ae. caballus* larvae were reported from Iranshahr County, located in the

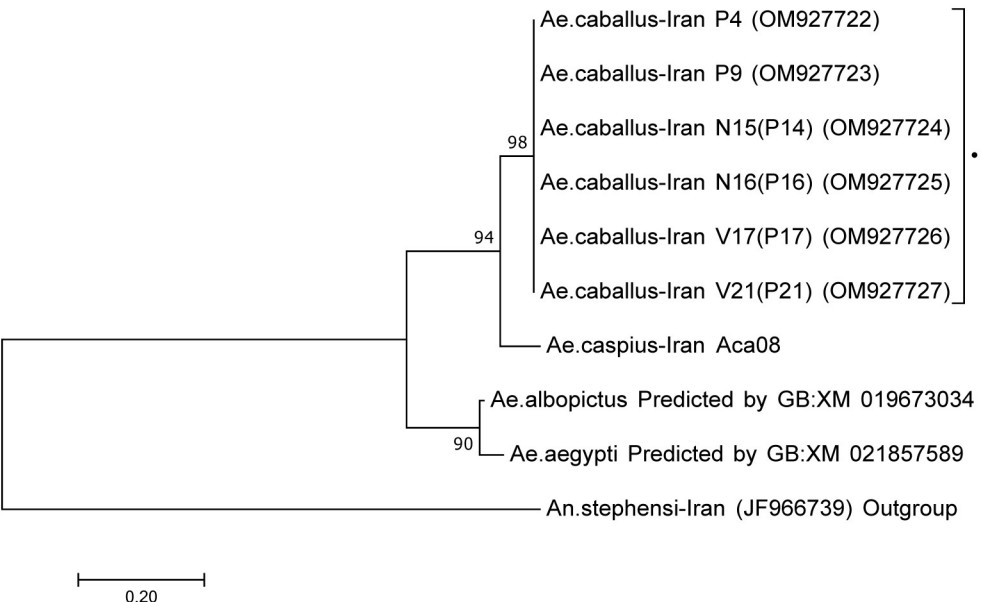

**Fig 8. Neighbour-joining tree, based on 340 bp DNA sequences of ANT fragment, created using MEGA7.** *Aedes caballus* from Iran are shown by bold triangle (▲), *Anopheles stephensi* used as an outgroup. Bootstrap values are shown on nodes. Scale bar shows genetic distance. Genbank IDs are shown on branches. Data were bootstrapped 500 times.

center of Sistan and Baluchistan Province [24]. In the most recent study conducted in this province, while no adult specimens of *Ae. caballus* were collected, a significant number of adult *Ae. vexans* (1465) were reported [32]. Based on the current results, the status of *Ae. vexans* is unclear and it may have been misidentified in previous studies. It's worth noting that the previous identification key for mosquitoes in Iran differs slightly to the most current Iranian key.

In the latest published key on Iranian mosquitoes, proboscis length is not regarded a character for identifying and distinguishing *Ae. caballus* from *Ae. vexans*, because it is considered longer than the forefemur in both *Ae. caballus Ae. vexans*. *Aedes vexans* is differentiated from *Ae. caballus* based on several characters, especially the ornamentation of abdominal terga [44]. In contrast, in the old key, *Ae. caballus* was keyed out, with *Ae. vittatus*, as species with short proboscis length than the forefemur [46]. That might lead to misidentification. Because, *Ae. caballus* has proboscis longer than the forefemur [44]. Misidentification of mosquitoes has been documented globally, highlighting the significance of accurate species identification, especially for vector-borne diseases [56]. So, correct identification employing validated keys and molecular analysis, when necessary, is crucial.

Heavy rains, typically associated with periodic monsoon systems, can create natural ponds or wetlands [57, 58]. *Aedes vexans* is known as a flood mosquito, often increases in population significantly following rainfall [59]. Surprisingly, in our study, *Ae. caballus* exhibited higher abundance. This suggests that *Ae. caballus* might also be monsoon-related.

Our findings align with collections made in Sudan and South Africa, where *Ae. caballus* was predominantly found in rural areas indicating a preference for this habitat. Both South Africa and our study utilized CO2-baited bednet and BG-Sentinel traps. Additionally, CDC miniature light traps and knock-down procedures were employed for *Aedes* collection in those countries [27, 60]. The bed net trap ($CO_2$/animal/human-baited) is widely used as an efficient collection method in the southeast of Iran Previous studies in this region have consistently reported the highest mosquitoes captures, including both anopheline or culicine species, using the bed net trap [42, 43, 61]. Our study similarly found that the bednet trap outperformed other trapping methods, highlighting its effectiveness for collecting *Ae. caballus*.

In light of the potential for misidentifying *Ae. caballus* in prior studies, we conducted DNA analysis to complement the morphological identification. Mitochondrial genes serve as valuable resources for such investigations, particularly when the morphological identification of species is challenging [62]. Our current results lead us to the conclusion that the *Ae. caballus* COI barcode region is distinctive to this species and does not exhibit close similarity with *Ae. vexans*.

Prior to the commencement of our study on May 25, 2018, no prior reference material for *Ae. caballus* regarding COI, ITS2 and ANT genes were available in the BOLD database or GenBank. In 2021, a published paper on Afrotropical mosquitoes included *Ae. caballus* accession numbers, enabling us to compare species from different regions based on partial COI nucleotide sequences [27]. However, we did not find any this species that involved ITS2 and ANT genes. Our phylogenetic analysis of COI gene revealed that African *Ae. caballus* formed a distinct sub-clade compared to our samples. It's worth noting that the accession length of the African samples ranged from 600 to 635 bp, with at least 40 bp missing when compared to our sequences. The utilization of GenBank samples required significant sequence trimming for all seven sequences, potentially reducing. overall variation. therefore, our COI sequence data may eventually prove more valuable for phylogenetic inference, especially when applied to larger sample sizes. While intraspecific variation within the published sequences was low ($\leq 0.5\% \pm 0.2$) [63], our results suggest the presence of different haplotypes in Iran based on COI and ITS2 genes. Further exploration of ITS2, which exhibited higher overall divergence compared

to COI, in *Ae. caballus* from various countries is warranted. The ANT gene effectively distinguished between the included *Aedes* species with strong bootstrap support and could serve as a valuable tool for resolving higher taxonomic relationships. The recommended best practice is the use of a combination of molecular markers such as ribosomal and mitochondrial, for resolving phylogenetic relationships [35, 64]. Given that divergence values for conspecifics are typically anticipated to be less than 0.5% [65], the observed genetic diversity of approximately 0.9% for COI in our studied samples underscores the necessity for further research. This research should encompass additional samples and locations, not limited to rural areas, and should include the temporal variation of specimens, with a particular focus on periods of heavy rainfall. Additionally, the divergence of $\geq 3.6\%$ between South African samples [63] and our specimens emphasizes the need for continued investigation.

## Conclusions

Accurate species identification is crucial for assessing disease threats posed by vector species. Our study highlights the importance of validated morphological keys and molecular identification. *Aedes caballus* appears to dominate this Oriental ecozone after heavy rainfall, making its potential disease transmission and control a subject of concern. We recommend retraining workshops and comprehensive entomological studies incorporating validated taxonomic keys and reliable molecular markers to enhance mosquito control efforts.

## Supporting information

**S1 Fig. Genetic diversity in *Aedes caballus* samples.** COI barcode region with 0.89% diversity, including 66.7% transitions and 33.3% transversions.
(PDF)

**S2 Fig. Genetic Diversity in of *Aedes caballus* samples.** ITS2 locus sequences with a diversity rate of 1.6%, characterized by six mutations, including two transitions and four transversions.
(PDF)

**S1 Raw images.**
(PDF)

## Acknowledgments

This research could not be accomplished without the assistance of our colleagues in Iranshahr health deputy. We are also grateful to Abdolmohsen Parvin, the head of CDC in Iranshahr University of Medical Sciences. We thank Farooq Askani for his assistance in field sampling and Eng. Hamid Reza Behjati in Sistan and Baluchistan Meteorological Organization for providing weather data.

## Author Contributions

**Conceptualization:** Jalil Nejati, Mohammad Ali Oshaghi.

**Data curation:** Jalil Nejati, Shahyad Azari-Hamidian, Vanessa L. White, Ary A. Hoffmann.

**Formal analysis:** Shahyad Azari-Hamidian, Ahmad A. Hanafi-Bojd, Fateh Karimian, Mona Koosha, Ary A. Hoffmann.

**Funding acquisition:** Mohammad Ali Oshaghi, Ary A. Hoffmann.

**Investigation:** Jalil Nejati, Vanessa L. White, Seyed H. Moosa-Kazemi.

**Methodology:** Mohammad Ali Oshaghi, Vanessa L. White, Rubén Bueno-Marí, Mona Koosha, Ary A. Hoffmann.

**Project administration:** Mohammad Ali Oshaghi, Ary A. Hoffmann.

**Resources:** Mohammad Ali Oshaghi, Ary A. Hoffmann.

**Software:** Vanessa L. White, Fateh Karimian.

**Supervision:** Hassan Vatandoost.

**Validation:** Mohammad Ali Oshaghi, Ary A. Hoffmann.

**Writing – original draft:** Jalil Nejati.

**Writing – review & editing:** Mohammad Ali Oshaghi, Vanessa L. White, Rubén Bueno-Marí, Nancy M. Endersby-Harshman, Jason K. Axford, Nayyereh Choubdar, Ary A. Hoffmann.

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
