## [Decision Letter · Decision Letter 0]

31 Aug 2023

PONE-D-23-24868The Monsoon-associated mosquito ‘Aedes caballus’; first record from southeastern Iran with a description of morphological, ecological, and molecular aspectsPLOS ONE

Dear Dr. Oshaghi,

Thank you for submitting your manuscript to PLOS ONE. After careful consideration, we feel that it has merit but does not fully meet PLOS ONE’s publication criteria as it currently stands. Therefore, we invite you to submit a revised version of the manuscript that addresses the points raised during the review process.

In particular, the reviewers felt that the manuscript needed further in depth descriptions of the species being studied. Additionally, please ensure that the revised manuscript also addresses the lack of detail on the sampling methodology as raised by Reviewer 1. Please also ensure that the other editorial comments raised by all three reviewers are addressed in the revised version you submit.

We look forward to receiving your revised manuscript.

Kind regards,

James Colborn

Academic Editor

PLOS ONE

Journal Requirements:

2. Template for MSs submitted from June 20, 2022 onwards (for reference only. Please use the text overlap send back provided by the staff editor in the PRTC notes)

Thank you for submitting the above manuscript to PLOS ONE. During our internal evaluation of the manuscript, we found significant text overlap between your submission and previous work in the [introduction, conclusion, etc.].

Please revise the manuscript to rephrase the duplicated text, cite your sources, and provide details as to how the current manuscript advances on previous work. Please note that further consideration is dependent on the submission of a manuscript that addresses these concerns about the overlap in text with published work.

[If the overlap is with the authors’ own works: Moreover, upon submission, authors must confirm that the manuscript, or any related manuscript, is not currently under consideration or accepted elsewhere. If related work has been submitted to PLOS ONE or elsewhere, authors must include a copy with the submitted article. Reviewers will be asked to comment on the overlap between related submissions (http://journals.plos.org/plosone/s/submission-guidelines#loc-related-manuscripts).]

We will carefully review your manuscript upon resubmission and further consideration of the manuscript is dependent on the text overlap being addressed in full. Please ensure that your revision is thorough as failure to address the concerns to our satisfaction may result in your submission not being considered further

3.Thank you for stating the following financial disclosure: 

5. Please amend your list of authors on the manuscript to ensure that each author is linked to an affiliation. Authors’ affiliations should reflect the institution where the work was done (if authors moved subsequently, you can also list the new affiliation stating “current affiliation:….” as necessary).

6. We note that [Figure 1] in your submission contain [map/satellite] images which may be copyrighted. All PLOS content is published under the Creative Commons Attribution License (CC BY 4.0), which means that the manuscript, images, and Supporting Information files will be freely available online, and any third party is permitted to access, download, copy, distribute, and use these materials in any way, even commercially, with proper attribution. For these reasons, we cannot publish previously copyrighted maps or satellite images created using proprietary data, such as Google software (Google Maps, Street View, and Earth). For more information, see our copyright guidelines: http://journals.plos.org/plosone/s/licenses-and-copyright.

A. You may seek permission from the original copyright holder of Figure 1] to publish the content specifically under the CC BY 4.0 license.  

B. If you are unable to obtain permission from the original copyright holder to publish these figures under the CC BY 4.0 license or if the copyright holder’s requirements are incompatible with the CC BY 4.0 license, please either i) remove the figure or ii) supply a replacement figure that complies with the CC BY 4.0 license. Please check copyright information on all replacement figures and update the figure caption with source information. If applicable, please specify in the figure caption text when a figure is similar but not identical to the original image and is therefore for illustrative purposes only.

Reviewers' comments:

Reviewer's Responses to Questions

**Comments to the Author**

1. Is the manuscript technically sound, and do the data support the conclusions?

Reviewer #1: Yes

Reviewer #2: Yes

Reviewer #3: Yes

2. Has the statistical analysis been performed appropriately and rigorously? 

Reviewer #1: N/A

Reviewer #2: N/A

Reviewer #3: N/A

3. Have the authors made all data underlying the findings in their manuscript fully available?

Reviewer #1: Yes

Reviewer #2: Yes

Reviewer #3: Yes

4. Is the manuscript presented in an intelligible fashion and written in standard English?

Reviewer #1: Yes

Reviewer #2: Yes

Reviewer #3: Yes

5. Review Comments to the Author

Reviewer #1: Dear authors,

The topic of this manuscript is very interesting and the results represent new data for the studied area. The manuscript is written in good and understandable English. Lack of the manuscript is knowledge during the study design. The authors are making parallel of Ae. caballus and Aedes vexans but they do not mention biology of these two species which is crucial for sampling. On the photos are given concrete, man-made breeding site and it is a question what species are using it? Is it used by vexans? Authors were giving wide spectrum of breeding sites but they cannot all be inhabited by vexans and maybe also not by caballus. So, it is crucial to explain the reasons why authors decided to do the sampling that way. Also, it would be important to know what the breeding site of the target species is and that could be known only if larvae were sampled.

Afterwards, my suggestion is to improve this MS significantly by adding the biology and comparison between these two species.

It is necessary to describe the locations selected for sampling. It is very imprecise to say just urban or rural.

It would be also useful to give the other species that were collected in traps.

There are also technical mistakes that should be corrected (please find it below) and incorrect citation.

Here are my specific comments.

What is ecological technique?

L53 Please correct to: “… are the most important insects from the public health point of view.” Or you can change it differently but not to call them medical arthropods.

L57 Major for what reason? This should be explained, changed or deleted.

L58-L60 It is not correct. Aedes mosquitoes exist in temperate climate as. It has to be deleted or changed.

L59-L60 Please delete word widespread because at the end of sentence you said around the world.

L62 Aedini or aedine?

L67 Please give the name of authors or the name of classifications in the text.

L69 Citation (9) is incorrect. Author should cite the publication what proved the vector competence of species and not the pictorial key.

L74 Citation (11) is also incorrect. It is reference about Aedes albopictus and authors need to cite some work about meteorological data in Iran for this sentence.

Figure 1b is not showing anything significant and it is very blurry. Should be deleted.

L75 Instead of saying in this region please give exact name of region.

L76 Aedes vexans is not vector of WNV. Please revise the literature about its vector competence. Also, the citation (12) is incorrect because it is about invasive mosquito species Aedes albopictus.

L80 It is called molecular identification.

L83 How do you mean “relative to”.

Figure 2c. It is not visible where the water is. Photo should be deleted or replaced with better one.

L117 Why not during the night? When is the targeted species active? Aedes vexans is active in the dusk and down so it would be adequate to set up traps overnight.

L121 Why every 15 min?

L129 Molecular analyses and not studies.

Table 1. Please delete “details of”

L183 Please delet subtitle because there could be only field sampling so the subtitle is redundant.

L184 Please correct to “In total, 1734 adults of Ae. caballus were collected”. Instead of saying “but only from rural sites” it is clearer when you say “All specimens were collected in rural sites”.

The authors gave a percentages and when you sum them it is 0.05 more the 100%. Please change to two decimals to avoid that.

Figure 3. Instead of Climatic variable as … should be written temperature (℃). Important question is how did the authors measure these parameter? Please specify that and give the source of information.

L199 Why first Fig 4c? What about a and b?

L283 Please explain why did you include Anopheles species in the comparison when it is completely different group of mosquitoes? They are very distinguishable from other mosquito genera.

L288 Please check the citation (29). It is not about vector competence.

L289 Why is relevant that species was not noted in two countries?

References: The references require formatizing. Latin names should be in italics, name of diseases with big letters etc. Please follow the propositions of the journal.

Reviewer #2: acceptance

the manuscript is good and introduce a new mosquito in southeast of iran

Considering the increase of vector-borne diseases in the southeast of Iran, the results of the study are a very good warning for the health centres of Iran and neighbouring countries to activate prevention measures.

Reviewer #3: The current study is a specialized, comprehensive study with valuable technical information. I am sure that its publication can be helpful to researchers in the fields of medical entomology and control of mosquito-borne diseases. Especially field experts involved in identifying important species of mosquitoes in Iran.

The following suggestions are provided to improve the current manuscript so that the authors can use them while paying attention to them:

In my opinion, the introduction of the present version needs to be rewritten in some parts. The following suggestions may help:

- It is better for the authors to provide a brief description of the morphological differences between the studied species (Ae. caballus) and the closely related species (Ae. vexans) (like what they have presented in the discussion section). It might even be better to move this section to the introduction.

- Although there are limitted studies (apparently only one article) about the molecular investigations of the studied species (Ae. caballus), it is better to refer to that article in the introduction as well as the justification of the markers used in this study.

A small recommendation: the writing of the name of the place (Parumi or Paroomi) should be standardized everywhere in the article (P4, L110, 112).

6. PLOS authors have the option to publish the peer review history of their article (what does this mean?). If published, this will include your full peer review and any attached files.

Reviewer #1: No

Reviewer #2: No

Reviewer #3: **Yes: **Alireza Chavshin

---

## [Author Response · Author response to Decision Letter 0]

19 Oct 2023

Dear respected Editor and Reviewers, 

Thank you for your constructive evaluation of our manuscript. The comments have been carefully considered, ‎‎and the manuscript was revised accordingly. We have made the requested corrections in the text point-by-point, which can be identified by the red words in the “Revised Manuscript with Track Changes”. In addition, based on the respected editor's comments, we made some edits to improve readability and eliminate duplicated words. We would like to express our gratitude once again to you, the editorial team, and the respected reviewers for your invaluable feedback.

Responses to Editor: 

please ensure that the revised manuscript also addresses the lack of detail on the sampling methodology as raised by Reviewer 1.

Answer: We endeavored to revise the version to address the details in the sampling method raised by reviewer 1. Furthermore, the response to his/her insightful question (line 117) can be found in the responses section. 

 It should be mentioened that:

• Here, you will find a rebuttal letter addressing the points raised by both you and the reviewer, labeled as “Response to Reviewers”.

• Additionally, we have included a marked-up copy of our manuscript that highlights the changes made to the original version, labeled “Revised Manuscript with Track Changes”. 

• You can also access an unmarked version without tracked changes, labeled simply as “Manuscript”.

Journal Requirements:

1. We have taken care to ensure that our manuscript adheres to PLOS ONE's style requirements, including file naming. It is based on the PLOS ONE style templates, especially aligns with the styles of published articles in this prestigious journal.

2. We have reviewed the references to ensure they conform to the PLOS style. These references were prepared using the EndNote style available on the PLOS website (https://endnote.com/style_download/plos-public-library-of-science-all-journals/).

We provided some edits to improve readability and eliminate duplicated words. We can confirm that neither this manuscript nor any related manuscript is currently under consideration or accepted elsewhere.

3. We have mentioned the sources of funding in the 'Acknowledgments' section. Since the funders had no role in the study , the sentence was stated “The funders had no role in study design, data collection and analysis, decision to publish, or preparation of the manuscript”. 

4. We have included the original, uncropped, and unadjusted gel result images in the Supporting Information files. It was stated in the Cover letter too.

5. The authors’ affiliations reflect the institution where the work was done. ‎

6. In Figure 1, we have included map images created using GIS software. Both images, the map and biting, were original and have not been previously published. Baesd on the the commnet of reviewer 1, we delete the Fig 1B, on biting. To address any potential copyright concerns regarding the map, we have added the following sentence as the legend of Fig 1 caption: “Prepared using ESRI ArcGIS 10.3 and Adobe Photoshop CS6 softwares”. 

7. We have revised all figures’ caption to adhere to the PLoS caption format.

8. The references were reviewed to ensure that they are complete and correct

Responses to Reviewer 1:

The topic of this manuscript is very interesting and the results represent new data for the studied area. The manuscript is written in good and understandable English. Lack of the manuscript is knowledge during the study design. The authors are making parallel of Ae. caballus and Aedes vexans but they do not mention biology of these two species which is crucial for sampling. On the photos are given concrete, man-made breeding site and it is a question what species are using it? Is it used by vexans? Authors were giving wide spectrum of breeding sites but they cannot all be inhabited by vexans and maybe also not by caballus. So, it is crucial to explain the reasons why authors decided to do the sampling that way. Also, it would be important to know what the breeding site of the target species is and that could be known only if larvae were sampled.

Afterwards, my suggestion is to improve this MS significantly by adding the biology and comparison between these two species.

Answer: We appreciate your positive remarks about the manuscript's topic, the novelty of the results, and the quality of the English language. We acknowledge your valuable observations regarding the manuscript's knowledge gap during study design, the need for more information on the biology of Aedes caballus and Aedes vexans, and the importance of explaining sampling approach. Your comments are well-received, and we will address these points in the revised manuscript to enhance its quality and clarity. Thank you for your constructive feedback.

- Thank you for your valuable suggestion regarding the biology and sampling. We have included information about the biology of these two species in the Introduction and provided additional details on the sampling process,, as you correctly pointed out.

‎-‎ Thank you for your attention to detail regarding the photos of the breeding sites. The intent ‎behind including these photos was to illustrate the different types of larval habitats in the study ‎area. However, it's important to note that our investigation focused solely on collecting adult ‎mosquitoes, and we did not examine larvae. You are absolutely right about the cement ‎man-made breeding site, and it was unrelated to these species' biology, which could confuse readers. Therefore, we have removed all the photos in Fig 2, replaced them with the adult collection traps, and modified the ‎caption. Thank you once again for your valuable input.‎

It is necessary to describe the locations selected for sampling. It is very imprecise to say just urban or ‎rural.

Answer: Thanks for your valuable comment. The selected locations for sampling‎ were described more in the subsection “study area” in the “Materials and methods“.

It would be also useful to give the other species that were collected in traps.

Answer: Thanks. Our study's primary focus was not to investigate and compare trap performance among different species. Therefore, we chose not to mention the names of other species, as doing so might divert the reader's attention from the main purpose of our study. 

There are also technical mistakes that should be corrected (please find it below) and incorrect citation.

Here are my specific comments.

What is ecological technique?

Answer: Thank you for your observation. ‎we means field observations and mosquito collection for sending to the laboratory. We have provided a brief explanation at the end of Introduction.

L53 Please correct to: “… are the most important insects from the public health point of view.” Or you can change it differently but not to call them medical arthropods.

Answer: Thank a lot. The recommended correction was performed.

L57 Major for what reason? This should be explained, changed or deleted.

Answer: I appreciate your accuracy.It was deleted.

L58-L60 It is not correct. Aedes mosquitoes exist in temperate climate as. It has to be deleted or changed.

Answer: Thank you. We removed the sentence based on your valuable comment as well as it didn't align with the paragraph's subject. 

L59-L60 Please delete word widespread because at the end of sentence you said around the world.

Answer: Thank you for your valuable suggestion. The sentence was removed.

L62 Aedini or aedine?

Answer: I appreciate your accuracy. It was corrected.

L67 Please give the name of authors or the name of classifications in the text.

Answer: The authors’ name and the references were added.

L69 Citation (9) is incorrect. Author should cite the publication what proved the vector competence of species and not the pictorial key.

Answer: In Huang's pictorial key, the medical importance of caballus is mentioned. However, in response to your valuable comment, we have incorporated more reliable references.

L74 Citation (11) is also incorrect. It is reference about Aedes albopictus and authors need to cite some work about meteorological data in Iran for this sentence.

Answer: Thank you for your recommendation. We have included references related to the southeastern climate of Iran.

Figure 1b is not showing anything significant and it is very blurry. Should be deleted.

Answer: It depicted Ae. caballus biting even through clothes, and we initially believed it would be useful for readers to learn more about its behavior. However, following your suggestion, we have removed it.

L75 Instead of saying in this region please give exact name of region.‎

Answer: Thanks, the correction was performed.

L76 Aedes vexans is not vector of WNV. Please revise the literature about its vector competence. Also, the citation (12) is incorrect because it is about invasive mosquito species Aedes albopictus.

Answer: I appreciate your accuracy. Based on laboratory studies, it appeared that vexans could ‎potentially play a role in the transmission of WNV. Following your valuable comment and more literature review, we knew you were right, so we deleted ‎the phrase " West Nile Viruses".

L80 It is called molecular identification.

Answer: Thanks a lot, the correction was performed.

L83 How do you mean “relative to”.

Answer: Thanks for your question. Our meaning was “in comparison to”. It was modified to avoid confusing the readers.

Figure 2c. It is not visible where the water is. Photo should be deleted or replaced with better one.

Answer: Thank you for your valuable feedback. While our intention with Figure 2 was to illustrate the environmental context of the study area, as you rightly pointed out, the images might have led readers to focus on the larval habitats, which were not the primary subject of our study. As a result, we have replaced all the photos in Figure 2 with images of the traps used for the sampling.

L117 Why not during the night? When is the targeted species active? Aedes vexans is active in the dusk and down so it would be adequate to set up traps overnight.

Answer: Thank you for your insightful question. Our study's primary objective did not involve comparing these two species’ prevalence or trap performance, which is why we did not continue the study into the morning. Furthermore, logistical challenges, including a shortage of vehicles and concerns about trap security, limited our ability to install the CDC light trap or other trap types for the entire night until sunrise to collect more samples of Ae.vexans. As you rightly pointed out, vexans exhibits crepuscular behavior and is most active at dusk. Therefore, in addition to daytime sampling, our efforts were concentrated on collecting specimens 1-2 hours after sunset. However, we mentioned the suitable traps and sampling times for Ae. vexans in the introduction. As the title suggests, our study primarily focuses on Ae. caballus. I apologize for the lengthy explanation and appreciate your understanding.

L121 Why every 15 min?

Answer: Thanks for your question. To prevent the possibility of mosquitoes escaping, particularly from the resting box trap, and to ensure the collection of all samples, we checked the traps every 15 minutes.

L129 Molecular analyses and not studies.

Answer: Thank you for your precision in correcting the word; it has been rectified.

Table 1. Please delete “details of”.

Answer: It was corrected. Thank you.

L183 Please delet subtitle because there could be only field sampling so the subtitle is redundant.

Answer: It was deleted.

L184 Please correct to “In total, 1734 adults of Ae. caballus were collected”. Instead of saying “but only from rural sites” it is clearer when you say “All specimens were collected in rural sites”.

Answer: Thanks a lot, the correction was performed.

The authors gave a percentages and when you sum them it is 0.05 more the 100%. Please change to ‎two decimals to avoid that.‎

Answer: I appreciate your accuracy. It was corrected.

Figure 3. Instead of Climatic variable as … should be written temperature (℃). Important question is how did the authors measure these parameter? Please specify that and give the source of information.

Answer: The vertical axis shows 2 meteorological variables; temperature and precipitation which we showed with ℃ and mm. The phrase “Climatic variable” was deleted . Based on your valuable comment, the source of meteorological information was added in the legend part of the Fig 3 caption.

L199 Why first Fig 4c? What about a and b?

Answer: I appreciate your accuracy. It was corrected.

L283 Please explain why did you include Anopheles species in the comparison when it is completely different group of mosquitoes? They are very distinguishable from other mosquito genera.

Answer: Thanks for your question. You are right. It was important for us to select a suitable outgroup of mosquitoes for the phylogenetic tree, and we believe Anopheles could fulfill this role.

L288 Please check the citation (29). It is not about vector competence.‎

Answer: Thank you for your precision. It has been removed.

L289 Why is relevant that species was not noted in two countries?

Answer: Thanks for your question. Since the southeast of Iran, Pakistan, and India share the Oriental ‎region with relatively similar species and based on ‎the literature review, caballus has not been reported from these 2 countries that may be interesting ‎for the readers. However, now, after your question, we think that it may be questionable for the readers, so we deleted that sentence.

References: The references require formatizing. Latin names should be in italics, name of diseases with ‎big letters etc. Please follow the propositions of the journal‎.

Answer: I appreciate your efforts in taking the time to review them. It was corrected based on the journal format in the “Manuscript without tracked changes”.

Response to Reviewer 2:

Reviewer #2: acceptance

the manuscript is good and introduce a new mosquito in southeast of iran considering the increase of vector-borne diseases in the southeast of Iran, the results of the study are a very good warning for the health centres of Iran and neighbouring countries to activate prevention measures

Answer: Thank you, for your positive feedback and acknowledgment of the significance of our study in introducing a new mosquito species in the southeast of Iran. We appreciate your recognition of the implications of our findings for public health and vector-borne disease prevention measures in Iran and neighboring regions. Your comments and support are greatly appreciated.

Responses to Reviewer 3:

Reviewer #3: The current study is a specialized, comprehensive study with valuable technical information. I am sure that its publication can be helpful to researchers in the fields of medical entomology and control of mosquito-borne diseases. Especially field experts involved in identifying important species of mosquitoes in Iran.

Answer: Thank you, for your positive assessment of our study. We are delighted to hear that you found it specialized, comprehensive, and valuable for researchers in the fields of medical entomology and mosquito-borne disease control. Your recognition of the potential benefits of our publication to field experts in identifying important mosquito species in Iran is greatly appreciated. Your feedback is invaluable to us.

The following suggestions are provided to improve the current manuscript so that the authors can use ‎them while paying attention to them: 

In my opinion, the introduction of the present version needs to be rewritten in some parts. The following ‎suggestions may help: 

‎- It is better for the authors to provide a brief description of the morphological differences between the ‎studied species (Ae. caballus) and the closely related species (Ae. vexans) (like what they have ‎presented in the discussion section). It might even be better to move this section to the introduction.

Answer: Thanks for your comment. Because our assumption is that the reader may not have detailed information about their morphological characteristics while reading the introduction, please let us to maintain the current approach and reserve th

---

## [Decision Letter · Decision Letter 1]

25 Jan 2024

The monsoon-associated equine South African pointy mosquito ‘Aedes caballus’; the first comprehensive record from southeastern Iran with a description of ecological, morphological, and molecular aspects

PONE-D-23-24868R1

Dear Dr. Oshaghi,

We’re pleased to inform you that your manuscript has been judged scientifically suitable for publication and will be formally accepted for publication once it meets all outstanding technical requirements.

Kind regards,

James Colborn

Academic Editor

PLOS ONE

Additional Editor Comments (optional):

Reviewers' comments:

Reviewer's Responses to Questions

**Comments to the Author**

1. If the authors have adequately addressed your comments raised in a previous round of review and you feel that this manuscript is now acceptable for publication, you may indicate that here to bypass the “Comments to the Author” section, enter your conflict of interest statement in the “Confidential to Editor” section, and submit your "Accept" recommendation.

Reviewer #2: All comments have been addressed

Reviewer #3: All comments have been addressed

2. Is the manuscript technically sound, and do the data support the conclusions?

Reviewer #2: Yes

Reviewer #3: Yes

3. Has the statistical analysis been performed appropriately and rigorously? 

Reviewer #2: Yes

Reviewer #3: N/A

4. Have the authors made all data underlying the findings in their manuscript fully available?

Reviewer #2: Yes

Reviewer #3: Yes

5. Is the manuscript presented in an intelligible fashion and written in standard English?

Reviewer #2: Yes

Reviewer #3: Yes

6. Review Comments to the Author

Reviewer #2: The topic of this manuscript is very interesting and the results represent new data for the studied area

Reviewer #3: Dear Authors,

Thank you for considering all of reviewers comments and revised your manuscript accordingly.

7. PLOS authors have the option to publish the peer review history of their article (what does this mean?). If published, this will include your full peer review and any attached files.

Reviewer #2: **Yes: **Hamzeh Alipour

Reviewer #3: No

---

## [Editor Report · Acceptance letter]

13 May 2024

PONE-D-23-24868R1 

PLOS ONE

Dear Dr. Oshaghi, 

I'm pleased to inform you that your manuscript has been deemed suitable for publication in PLOS ONE. Congratulations! Your manuscript is now being handed over to our production team.

Kind regards, 

on behalf of

Dr. James Colborn 

Academic Editor

PLOS ONE